# Provably Efficient Reinforcement Learning with Linear Function Approximation under Adaptivity Constraints

**Tianhao Wang**[*]
Department of Statistics and Data Science
Yale University
New Haven, CT 06511
`tianhao.wang@yale.edu`

**Dongruo Zhou**[*]
Department of Computer Science
University of California, Los Angeles
Los Angeles, CA 90095
`drzhou@cs.ucla.edu`

**Quanquan Gu**
Department of Computer Science
University of California, Los Angeles
Los Angeles, CA 90095
`qgu@cs.ucla.edu`

## Abstract

We study reinforcement learning (RL) with linear function approximation under the adaptivity constraint. We consider two popular limited adaptivity models: the batch learning model and the rare policy switch model, and propose two efficient online RL algorithms for episodic linear Markov decision processes, where the transition probability and the reward function can be represented as a linear function of some known feature mapping. In specific, for the batch learning model, our proposed LSVI-UCB-Batch algorithm achieves an $\widetilde{O}(\sqrt{d^3 H^3 T} + dHT/B)$ regret, where $d$ is the dimension of the feature mapping, $H$ is the episode length, $T$ is the number of interactions and $B$ is the number of batches. Our result suggests that it suffices to use only $\sqrt{T/dH}$ batches to obtain $\widetilde{O}(\sqrt{d^3 H^3 T})$ regret. For the rare policy switch model, our proposed LSVI-UCB-RareSwitch algorithm enjoys an $\widetilde{O}(\sqrt{d^3 H^3 T[1 + T/(dH)]^{dH/B}})$ regret, which implies that $dH \log T$ policy switches suffice to obtain the $\widetilde{O}(\sqrt{d^3 H^3 T})$ regret. Our algorithms achieve the same regret as the LSVI-UCB algorithm (Jin et al., 2020), yet with a substantially smaller amount of adaptivity. We also establish a lower bound for the batch learning model, which suggests that the dependency on $B$ in our regret bound is tight.

## 1 Introduction

Real-world reinforcement learning (RL) applications often come with possibly infinite state and action space, and in such a situation classical RL algorithms developed in the tabular setting are not applicable anymore. A popular approach to overcoming this issue is by applying function approximation techniques to the underlying structures of the Markov decision processes (MDPs). For example, one can assume that the transition probability and the reward are linear functions of a known feature mapping $\phi : \mathcal{S} \times \mathcal{A} \to \mathbb{R}^d$, where $\mathcal{S}$ and $\mathcal{A}$ are the state space and action space, and $d$ is the dimension of the embedding. This gives rise to the so-called linear MDP model (Yang and Wang, 2019; Jin et al., 2020). Assuming access to a generative model, efficient algorithms under

---

[*]Equal contribution

35th Conference on Neural Information Processing Systems (NeurIPS 2021).

this setting have been proposed by Yang and Wang (2019) and Lattimore et al. (2020). For online finite-horizon episodic linear MDPs, Jin et al. (2020) proposed an LSVI-UCB algorithm that achieves $\widetilde{O}(\sqrt{d^3 H^3 T})$ regret, where $H$ is the planning horizon (i.e., length of each episode) and $T$ is the number of interactions.

However, all the aforementioned algorithms require the agent to update the policy in every episode. In practice, it is often unrealistic to frequently switch the policy in the face of big data, limited computing resources as well as inevitable switching costs. Thus one may want to batch the data stream and update the policy at the end of each period. For example, in clinical trials, each phase (batch) of the trial amounts to applying a medical treatment to a batch of patients in parallel. The outcomes of the treatment are not observed until the end of the phase and will be subsequently used to design experiments for the next phase. Choosing the appropriate number and sizes of the batches is crucial to achieving nearly optimal efficiency for the clinical trial. This gives rise to the limited adaptivity setting, which has been extensively studied in many online learning problems including prediction-from-experts (PFE) (Kalai and Vempala, 2005; Cesa-Bianchi et al., 2013), multi-armed bandits (MAB) (Arora et al., 2012; Cesa-Bianchi et al., 2013) and online convex optimization (Jaghargh et al., 2019; Chen et al., 2020), to mention a few. Nevertheless, in the RL setting, learning with limited adaptivity is relatively less studied. Bai et al. (2019) introduced two notions of adaptivity in RL, *local switching cost* and *global switching cost*, that are defined as follows

$$N_{\text{local}} = \sum_{k=1}^{K-1} \sum_{h=1}^{H} \sum_{s \in \mathcal{S}} \mathbb{1}\{\pi_h^k(s) \neq \pi_h^{k+1}(s)\} \quad \text{and} \quad N_{\text{global}} = \sum_{k=1}^{K-1} \mathbb{1}\{\pi^k \neq \pi^{k+1}\}, \qquad (1.1)$$

where $\pi^k = \{\pi_h^k : \mathcal{S} \to \mathcal{A}\}_{h \in [H]}$ is the policy for the $k$-th episode of the MDP, $\pi^k \neq \pi^{k+1}$ means that there exists some $(h, s) \in [H] \times \mathcal{S}$ such that $\pi_h^k(s) \neq \pi_h^{k+1}(s)$, and $K$ is the number of episodes. Then they proposed a Q-learning method with UCB2H exploration that achieves $\widetilde{O}(\sqrt{H^3 SAT})$ regret with $O(H^3 SA \log(T/(AH)))$ local switching cost for tabular MDPs, but they did not provide tight bounds on the global switching cost.

In this paper, based on the above motivation, we aim to develop online RL algorithms with linear function approximation under adaptivity constraints. In detail, we consider time-inhomogeneous[2] episodic linear MDPs (Jin et al., 2020) where both the transition probability and the reward function are unknown to the agent. In terms of the limited adaptivity imposed on the agent, we consider two scenarios that have been previously studied in the online learning literature (Perchet et al., 2016; Abbasi-Yadkori et al., 2011): the batch learning model and the rare policy switch model. More specifically, in the batch learning model (Perchet et al., 2016), the agent is forced to pre-determine the number of batches (or equivalently batch size). Within each batch, the same policy is used to select actions, and the policy is updated only at the end of this batch. The amount of adaptivity in the batch learning model is measured by the number of batches, which is expected to be as small as possible. In contrast, in the rare policy switch model (Abbasi-Yadkori et al., 2011), the agent can adaptively choose when to switch the policy and therefore start a new batch in the learning process as long as the total number of policy updates does not exceed the given budget on the number of policy switches. The amount of adaptivity in the rare policy switch model can be measured by the number of policy switches, which turns out to be the same as the global switching cost introduced in Bai et al. (2019). It is worth noting that for the same amount of adaptivity[3], the rare policy switch model can be seen as a relaxation of the batch learning model since the agent in the batch learning model can only change the policy at pre-defined time steps. In our work, for each of these limited adaptivity models, we propose a variant of the LSVI-UCB algorithm (Jin et al., 2020), which can be viewed as an RL algorithm with full adaptivity in the sense that it switches the policy at a per-episode scale. Our algorithms can attain the same regret as LSVI-UCB, yet with a substantially smaller number of batches/policy switches. This enables parallel learning and improves the large-scale deployment of RL algorithms with linear function approximation.

The main contributions of this paper are summarized as follows:

---

[2]We say an episodic MDP is time-inhomogeneous if its reward and transition probability are different at different stages within each episode. See Definition 3.2 for details.

[3]The number of batches in the batch learning model is comparable to the number of policy switches in the rare policy switch model.

- For the *batch learning* model, we propose an LSVI-UCB-Batch algorithm for linear MDPs and show that it enjoys an $\widetilde{O}(\sqrt{d^3 H^3 T} + dHT/B)$ regret, where $d$ is the dimension of the feature mapping, $H$ is the episode length, $T$ is the number of interactions and $B$ is the number of batches. Our result suggests that it suffices to use only $\sqrt{T/dH}$ batches, rather than $T$ batches, to obtain the same regret $\widetilde{O}(\sqrt{d^3 H^3 T})$ achieved by LSVI-UCB (Jin et al., 2020) in the fully sequential decision model. We also prove a lower bound of the regret for this model, which suggests that the required number of batches $\widetilde{O}(\sqrt{T})$ is sharp.

- For the *rare policy switch* model, we propose an LSVI-UCB-RareSwitch algorithm for linear MDPs and show that it enjoys an $\widetilde{O}(\sqrt{d^3 H^3 T[1 + T/(dH)]^{dH/B}})$ regret, where $B$ is the number of policy switches. Our result implies that $dH \log T$ policy switches are sufficient to obtain the same regret $\widetilde{O}(\sqrt{d^3 H^3 T})$ achieved by LSVI-UCB. The number of policy switches is much smaller than that[4] of the batch learning model when $T$ is large.

Concurrent to our work, Gao et al. (2021) proposed an algorithm achieving $\widetilde{O}(\sqrt{d^3 H^3 T})$ regret with a $O(dH \log K)$ global switching cost in the rare policy switch model. They also proved a $\Omega(dH/\log d)$ lower bound on the global switching cost. The focus of our paper is different from theirs: our goal is to design efficient RL algorithms under a switching cost budget $B$, while their goal is to achieve the optimal rate in terms of $T$ with as little switching cost as possible. On the other hand, for the rare policy switch model, our proposed algorithm (LSVI-UCB-RareSwitch) along its regret bound can imply their results by optimizing our regret bound concerning the switching cost budget $B$.

The rest of the paper is organized as follows. In Section 2 we discuss previous works related to this paper, with a focus on RL with linear function approximation and online learning with limited adaptivity. In Section 3 we introduce necessary preliminaries for MDPs and adaptivity constraints. Sections 4 and 5 present our proposed algorithms and the corresponding theoretical results for the batch learning model and the rare policy switch model respectively. In Section 6 we present the numerical experiment which supports our theory. Finally, we conclude our paper and point out a future direction in Section 7.

**Notation** We use lower case letters to denote scalars and use lower and upper case boldface letters to denote vectors and matrices respectively. For any real number $a$, we write $[a]^+ = \max(a, 0)$. For a vector $\mathbf{x} \in \mathbb{R}^d$ and matrix $\mathbf{\Sigma} \in \mathbb{R}^{d \times d}$, we denote by $\|\mathbf{x}\|_2$ the Euclidean norm and define $\|\mathbf{x}\|_{\mathbf{\Sigma}} = \sqrt{\mathbf{x}^\top \mathbf{\Sigma} \mathbf{x}}$. For any positive integer $n$, we denote by $[n]$ the set $\{1, \dots, n\}$. For any finite set $A$, we denote by $|A|$ the cardinality of $A$. For two sequences $\{a_n\}$ and $\{b_n\}$, we write $a_n = O(b_n)$ if there exists an absolute constant $C$ such that $a_n \leq C b_n$, and we write $a_n = \Omega(b_n)$ if there exists an absolute constant $C$ such that $a_n \geq C b_n$. We use $\widetilde{O}(\cdot)$ to further hide the logarithmic factors.

## 2 Related Works

**Reinforcement Learning with Linear Function Approximation** Recently, there have been many advances in RL with function approximation, especially the linear case. Jin et al. (2020) proposed an efficient algorithm for the first time for linear MDPs of which the transition probability and the rewards are both linear functions with respect to a feature mapping $\phi : \mathcal{S} \times \mathcal{A} \to \mathbb{R}^d$. Under similar assumptions, different settings (e.g., discounted MDPs) have also been studied in Yang and Wang (2019); Du et al. (2020); Zanette et al. (2020); Neu and Pike-Burke (2020) and He et al. (2021). A parallel line of work studies linear mixture MDPs (a.k.a. linear kernel MDPs) based on a ternary feature mapping $\psi : \mathcal{S} \times \mathcal{A} \times \mathcal{S} \to \mathbb{R}^d$ (see Jia et al. (2020); Zhou et al. (2021b); Cai et al. (2020); Zhou et al. (2021a)). For other function approximation settings, we refer readers to generalized linear model (Wang et al., 2021), general function approximation with Eluder dimension (Wang et al., 2020; Ayoub et al., 2020), kernel approximation (Yang et al., 2020), function approximation with disagreement coefficients (Foster et al., 2021) and bilinear classes (Du et al., 2021).

**Online Learning with Limited Adaptivity** As we mentioned before, online learning with limited adaptivity has been studied in two popular models of adaptivity constraints: the batch learning model and the rare policy switch model.

---

[4]The number of policy switches is identical to the number of batches in the batch learning model.

For the *batch learning model*, Altschuler and Talwar (2018) proved that the optimal regret bound for prediction-from-experts (PFE) is $\widetilde{O}(\sqrt{T \log n})$ when the number of batches $B = \Omega(\sqrt{T \log n})$, and $\min(\widetilde{O}(T \log n/B), T)$ when $B = O(\sqrt{T \log n})$, exhibiting a phase-transition phenomenon[5]. Here $T$ is the number of rounds and $n$ is the number of actions. For general online convex optimization, Chen et al. (2020) showed that the minimax regret bound is $\widetilde{O}(T/\sqrt{B})$. Perchet et al. (2016) studied batched 2-arm bandits, and Gao et al. (2019) studied the batched multi-armed bandits (MAB). Dekel et al. (2014) proved a $\Omega(T/\sqrt{B})$ lower bound for batched MAB, and Altschuler and Talwar (2018) further characterized the dependence on the number of actions $n$ and showed that the corresponding minimax regret bound is $\min(\widetilde{O}(T\sqrt{n}/\sqrt{B}), T)$. For batched linear bandits with adversarial contexts, Han et al. (2020) showed that the minimax regret bound is $\widetilde{O}(\sqrt{dT} + dT/B)$ where $d$ is the dimension of the context vectors. Better rates can be achieved for batched linear bandits with stochastic contexts as shown in Esfandiari et al. (2021); Han et al. (2020); Ruan et al. (2020).

For the *rare policy switch model*, the minimax optimal regret bound for PFE is $O(\sqrt{T \log n})$ in terms of both the expected regret (Kalai and Vempala, 2005; Geulen et al., 2010; Cesa-Bianchi et al., 2013; Devroye et al., 2015) and high-probability guarantees (Altschuler and Talwar, 2018), where $T$ is the number of rounds, and $n$ is the number of possible actions. For MAB, the minimax regret bound has been shown to be $\widetilde{O}(T^{2/3}n^{1/3})$ by Arora et al. (2012); Dekel et al. (2014). For stochastic linear bandits, Abbasi-Yadkori et al. (2011) proposed a rarely switching OFUL algorithm achieving $\widetilde{O}(d\sqrt{T})$ regret with $\log(T)$ batches. Ruan et al. (2020) proposed an algorithm achieving $\widetilde{O}(\sqrt{dT})$ regret with less than $O(d \log d \log T)$ batches for stochastic linear bandits with adversarial contexts.

For episodic RL with finite state and action space, Bai et al. (2019) proposed an algorithm achieving $\widetilde{O}(\sqrt{H^3 SAT})$ regret with $O(H^3 SA \log(T/(AH)))$ local switching cost where $S$ and $A$ are the number of states and actions respectively. They also provided a $\Omega(HSA)$ lower bound on the local switching cost that is necessary for sublinear regret. For the global switching cost, Zhang et al. (2021) proposed an MVP algorithm with at most $O(SA \log(KH))$ global switching cost for time-homogeneous tabular MDPs.

## 3  Preliminaries

### 3.1  Markov Decision Processes

We consider the time-inhomogeneous episodic Markov decision process, which is denoted by a tuple $M(\mathcal{S}, \mathcal{A}, H, \{r_h\}_{h \in [H]}, \{\mathbb{P}_h\}_{h \in [H]})$. Here $\mathcal{S}$ is the state space (may be infinite), $\mathcal{A}$ is the action space where we allow the feasible action set to change from step to step, $H$ is the length of each episode, and $r_h : \mathcal{S} \times \mathcal{A} \to [0, 1]$ is the reward function for each stage $h \in [H]$. At each stage $h \in [H]$, $\mathbb{P}_h(s'|s, a)$ is the transition probability function which represents the probability for state $s$ to transit to state $s'$ given action $a$. A policy $\pi$ consists of $H$ mappings, $\{\pi_h : \mathcal{S} \to \mathcal{A}\}_{h \in [H]}$. For any policy $\pi$, we define the action-value function $Q_h^\pi(s, a)$ and value function $V_h^\pi(s)$ as follows:

$$Q_h^\pi(s, a) = r_h(s, a) + \mathbb{E}_\pi\left[ \sum_{i=h}^{H} r_i(s_i, a_i) \Big| s_h = s, a_h = a \right], \qquad V_h^\pi(s) = Q_h^\pi(s, \pi_h(s)),$$

where $a_i \sim \pi_i(\cdot|s_i)$ and $s_{i+1} \sim \mathbb{P}_i(\cdot|s_i, a_i)$. The optimal value function $V_h^*$ and the optimal action-value function $Q_h^*(s, a)$ are defined as $V^*(s) = \sup_\pi V_h^\pi(s)$ and $Q_h^*(s, a) = \sup_\pi Q_h^\pi(s, a)$, respectively. For simplicity, for any function $V : \mathcal{S} \to \mathbb{R}$, we denote $[\mathbb{P}V](s, a) = \mathbb{E}_{s' \sim \mathbb{P}(\cdot|s, a)} V(s')$. In the online learning setting, at the beginning of $k$-th episode, the agent chooses a policy $\pi^k$ and the environment selects an initial state $s_1^k$, then the agent interacts with environment following policy $\pi^k$ and receives states $s_h^k$ and rewards $r_h(s_h^k, a_h^k)$ for $h \in [H]$. To measure the performance of the algorithm, we adopt the following notion of the total regret, which is the summation of suboptimalities between policy $\pi^k$ and optimal policy $\pi^*$:

**Definition 3.1.** We denote $T = KH$, and the regret Regret($T$) is defined as

$$\text{Regret}(T) = \sum_{k=1}^{K} \left[ V_1^*(s_1^k) - V_1^{\pi^k}(s_1^k) \right].$$

---

[5]They call it $B$-switching budget setting, which is identical to the batch learning model.

## 3.2 Linear Function Approximation

In this work, we consider a special class of MDPs called *linear MDPs* (Yang and Wang, 2019; Jin et al., 2020), where both the transition probability function and reward function can be represented as a linear function of a given feature mapping $\phi : \mathcal{S} \times \mathcal{A} \to \mathbb{R}^d$. Formally speaking, we have the following definition for linear MDPs.

**Definition 3.2.** $M(\mathcal{S}, \mathcal{A}, H, \{r_h\}_{h \in [H]}, \{\mathbb{P}_h\}_{h \in [H]})$ is called a linear MDP if there exist a *known* feature mapping $\phi(s, a) : \mathcal{S} \times \mathcal{A} \to \mathbb{R}^d$, *unknown* measures $\{\boldsymbol{\mu}_h = (\mu_h^{(1)}, \cdots, \mu_h^{(d)})\}_{h \in [H]}$ over $\mathcal{S}$ and unknown vectors $\{\boldsymbol{\theta}_h \in \mathbb{R}^d\}_{h \in [H]}$ with $\max_{h \in [H]}\{\|\boldsymbol{\mu}_h(\mathcal{S})\|_2, \|\boldsymbol{\theta}_h\|\} \leq \sqrt{d}$, such that the following holds for all $h \in [H]$:

- For any state-action-state triplet $(s, a, s') \in \mathcal{S} \times \mathcal{A} \times \mathcal{S}$, $\mathbb{P}_h(s'|s, a) = \langle \phi(s, a), \boldsymbol{\mu}_h(s') \rangle$.

- For any state-action pair $(s, a) \in \mathcal{S} \times \mathcal{A}$, $r_h(s, a) = \langle \phi(s, a), \boldsymbol{\theta}_h \rangle$.

Without loss of generality, we also assume that $\|\phi(s, a)\|_2 \leq 1$ for all $(s, a) \in \mathcal{S} \times \mathcal{A}$.

With Definition 3.2, it is shown in Jin et al. (2020) that the action-value function can be written as a linear function of the features.

**Proposition 3.3** (Proposition 2.3, Jin et al. 2020). For a linear MDP, for any policy $\pi$, there exist weight vectors $\{\mathbf{w}_h^\pi\}_{h \in [H]}$ such that for any $(s, a, h) \in \mathcal{S} \times \mathcal{A} \times [H]$, we have $Q_h^\pi(s, a) = \langle \phi(s, a), \mathbf{w}_h^\pi \rangle$. Moreover, we have $\|\mathbf{w}_h^\pi\|_2 \leq 2H\sqrt{d}$ for all $h \in [H]$.

Therefore, with the known feature mapping $\phi(\cdot, \cdot)$, it suffices to estimate the weight vectors $\{\mathbf{w}_h^\pi\}_{h \in [H]}$ in order to recover the action-value functions. This is the core idea behind almost all the algorithms and theoretical analyses for linear MDPs.

## 3.3 Models for Limited Adaptivity

In this work, we consider RL algorithms with limited adaptivity. There are two typical models for online learning with such limited adaptivity: *batch learning model* (Perchet et al., 2016) and *rare policy switch model* (Abbasi-Yadkori et al., 2011).

For the batch learning model, the agent pre-determines the batch grids $1 = t_1 < t_2 < \cdots < t_B < t_{B+1} = K + 1$ at the beginning of the algorithm, where $B$ is the number of batches. The $b$-th batch consists of $t_b$-th to $(t_{b+1} - 1)$-th episodes, and the agent follows the same policy within each batch. The adaptivity is measured by the number of batches.

For the rare policy switch model, the agent can decide whether she wants to switch the current policy or not. The adaptivity is measured by the number of policy switches, which is defined as

$$N_{\text{switch}} = \sum_{k=1}^{K-1} \mathbb{1}\{\pi^k \neq \pi^{k+1}\},$$

where $\pi^k \neq \pi^{k+1}$ means that there exists some $(h, s) \in [H] \times \mathcal{S}$ such that $\pi_h^k(s) \neq \pi_h^{k+1}(s)$. It is worth noting that $N_{\text{switch}}$ is identical to the *global switching cost* defined in (1.1).

Given a budget on the number of batches or the number of policy switches, we aim to design RL algorithms with linear function approximation that can achieve the same regret as their full adaptivity counterpart, e.g., LSVI-UCB (Jin et al., 2020).

# 4 RL in the Batch Learning Model

In this section, we consider RL with linear function approximation in the batch learning model, where given the number of batches $B$, we need to pin down the batches before the agent starts to interact with the environment.

## 4.1 Algorithm and Regret Analysis

We propose LSVI-UCB-Batch algorithm as displayed in Algorithm 1, which can be regarded as a variant of the LSVI-UCB algorithm proposed in Jin et al. (2020) yet with limited adaptivity.

---

**Algorithm 1** LSVI-UCB-Batch

---

**Require:** Number of batches $B$, confidence radius $\beta$, regularization parameter $\lambda$

1: Set $b \leftarrow 1$, $t_i \leftarrow (i-1) \cdot \lfloor K/B \rfloor + 1, i \in [B]$
2: **for** episode $k = 1, 2, \ldots, K$ **do**
3:     Receive the initial state $s_1^k$
4:     **if** $k = t_b$ **then**
5:         $b \leftarrow b + 1$, $Q_{H+1}^k(\cdot, \cdot) \leftarrow 0$
6:         **for** stage $h = H, H-1, \ldots, 1$ **do**
7:             $\mathbf{\Lambda}_h^k \leftarrow \sum_{\tau=1}^{k-1} \boldsymbol{\phi}(s_h^\tau, a_h^\tau) \boldsymbol{\phi}(s_h^\tau, a_h^\tau)^\top + \lambda \mathbf{I}$
8:             $\mathbf{w}_h^k \leftarrow (\mathbf{\Lambda}_h^k)^{-1} \sum_{\tau=1}^{k-1} \boldsymbol{\phi}(s_h^\tau, a_h^\tau) \cdot [r_h(s_h^\tau, a_h^\tau) + \max_{a \in \mathcal{A}} Q_{h+1}^k(s_{h+1}^\tau, a)]$
9:             $\Gamma_h^k(\cdot, \cdot) \leftarrow \beta \cdot [\boldsymbol{\phi}(\cdot, \cdot)^\top (\mathbf{\Lambda}_h^k)^{-1} \boldsymbol{\phi}(\cdot, \cdot)]^{1/2}$
10:            $Q_h^k(\cdot, \cdot) \leftarrow \min\{\boldsymbol{\phi}(\cdot, \cdot)^\top \mathbf{w}_h^k + \Gamma_h^k(\cdot, \cdot), H-h+1\}^+$, $\pi_h^k(\cdot) \leftarrow \operatorname{argmax}_{a \in \mathcal{A}} Q_h^k(\cdot, a)$
11:         **end for**
12:     **else**
13:         $Q_h^k \leftarrow Q_h^{k-1}$, $\pi_h^k \leftarrow \pi_h^{k-1}, \forall h \in [H]$
14:     **end if**
15:     **for** stage $h = 1 \ldots, H$ **do**
16:         Take the action $a_h^k \leftarrow \pi_h^k(s_h^k)$, receive the reward $r_h(s_h^k, a_h^k)$ and the next state $s_{h+1}^k$
17:     **end for**
18: **end for**

---

Algorithm 1 takes a series of batch grids $\{t_1, \ldots, t_{B+1}\}$ as input, where the $i$-th batch starts at $t_i$ and ends at $t_{i+1} - 1$. LSVI-UCB-Batch takes the uniform batch grids as its selection of grids, i.e., $t_i = (i-1) \cdot \lfloor K/B \rfloor + 1, i \in [B]$. By Proposition 3.3, we know that for each $h \in [H]$, the optimal value function $Q_h^*$ has the linear form $\langle \boldsymbol{\phi}(\cdot, \cdot), \mathbf{w}_h^* \rangle$. Therefore, to estimate the $Q_h^*$, it suffices to estimate $\mathbf{w}_h^*$. At the beginning of each batch, Algorithm 1 calculates $\mathbf{w}_h^k$ as an estimate of $\mathbf{w}_h^*$ by ridge regression (Line 8). Meanwhile, in order to measure the uncertainty of $\mathbf{w}_h^k$, Algorithm 1 sets the estimate $Q_h^k(\cdot, \cdot)$ as the summation of the linear function $\langle \boldsymbol{\phi}(\cdot, \cdot), \mathbf{w}_h^k \rangle$ and a Hoeffding-type exploration bonus term $\Gamma_h^k(\cdot, \cdot)$ (Line 10), which is calculated based on the confidence radius $\beta$. Then it sets the policy $\pi_h^k$ as the greedy policy with respect to $Q_h^k$. Within each batch, Algorithm 1 simply keeps the policy used in the previous episode without updating (Line 13). Apparently, the number of batches of Algorithm 1 is $B$.

Here we would like to make a comparison between our LSVI-UCB-Batch and other related algorithms. The most related algorithm is LSVI-UCB proposed in Jin et al. (2020). The main difference between LSVI-UCB-Batch and LSVI-UCB is the introduction of batches. In detail, when $B = K$, LSVI-UCB-Batch degenerates to LSVI-UCB. Another related algorithm is the SBUCB algorithm proposed by Han et al. (2020). Both LSVI-UCB-Batch and SBUCB take uniform batch grids as the selection of batches. The difference is that SBUCB is designed for linear bandits, which is a special case of episodic MDPs with $H = 1$.

The following theorem presents the regret bound of Algorithm 1.

**Theorem 4.1.** There exists a constant $c > 0$ such that for any $\delta \in (0, 1)$, if we set $\lambda = 1$, $\beta = cdH\sqrt{\log(2dT/\delta)}$, then under Assumption 3.2, the total regret of Algorithm 1 is bounded by

$$\text{Regret}(T) \leq 2H\sqrt{T \log\left(\frac{2dT}{\delta}\right)} + \frac{dHT}{2B \log 2} \log\left(\frac{T}{dH} + 1\right)$$
$$+ 4c\sqrt{2d^3 H^3 T \log\left(\frac{2dT}{\delta}\right) \log\left(\frac{T}{dH} + 1\right)}$$

with probability at least $1 - \delta$.

Theorem 4.1 suggests that the total regret of Algorithm 1 is bounded by $\widetilde{O}(\sqrt{d^3 H^3 T} + dHT/B)$. When $B = \Omega(\sqrt{T/dH})$, the regret of Algorithm 1 is $\widetilde{O}(\sqrt{d^3 H^3 T})$, which is the same as that of LSVI-UCB in Jin et al. (2020). However, it is worth noting that LSVI-UCB needs $K$ batches, while Algorithm 1 only requires $\sqrt{T/dH}$ batches, which can be much smaller than $K$.

**Algorithm 2** LSVI-UCB-RareSwitch
___
**Require:** Policy switch parameter $\eta$, confidence radius $\beta$, regularization parameter $\lambda$
1: Initialize $\mathbf{\Lambda}_h = \mathbf{\Lambda}_h^0 = \lambda \mathbf{I}_d$ for all $h \in [H]$
2: **for** episode $k = 1, 2, \dots, K$ **do**
3:      Receive the initial state $s_1^k$
4:      **for** stage $h = 1, 2, \cdots, H$ **do**
5:         $\mathbf{\Lambda}_h^k \leftarrow \sum_{\tau=1}^{k-1} \phi(s_h^\tau, a_h^\tau) \phi(s_h^\tau, a_h^\tau)^\top + \lambda \mathbf{I}_d$
6:      **end for**
7:      **if** $\exists h \in [H], \det(\mathbf{\Lambda}_h^k) > \eta \cdot \det(\mathbf{\Lambda}_h)$ **then**
8:         $Q_{H+1}^k(\cdot, \cdot) \leftarrow 0$
9:         **for** step $h = H, H-1, \cdots, 1$ **do**
10:           $\mathbf{\Lambda}_h \leftarrow \mathbf{\Lambda}_h^k$
11:           $\mathbf{w}_h^k \leftarrow (\mathbf{\Lambda}_h^k)^{-1} \sum_{\tau=1}^{k-1} \phi(s_h^\tau, a_h^\tau) \cdot [r_h(s_h^\tau, a_h^\tau) + \max_{a \in \mathcal{A}} Q_{h+1}^k(s_{h+1}^\tau, a)]$
12:           $\Gamma_h^k(\cdot, \cdot) \leftarrow \beta \cdot [\phi(\cdot, \cdot)^\top (\mathbf{\Lambda}_h^k)^{-1} \phi(\cdot, \cdot)]^{1/2}$
13:           $Q_h^k(\cdot, \cdot) \leftarrow \min\{\phi(\cdot, \cdot)^\top \mathbf{w}_h^k + \Gamma_h^k(\cdot, \cdot), H - h + 1\}^+, \pi_h^k(\cdot) \leftarrow \mathrm{argmax}_{a \in \mathcal{A}} Q_h^k(\cdot, a)$
14:      **end for**
15:      **else**
16:         $Q_h^k \leftarrow Q_h^{k-1}, \pi_h^k \leftarrow \pi_h^{k-1}, \forall h \in [H]$
17:      **end if**
18:      **for** stage $h = 1 \dots, H$ **do**
19:         Take the action $a_h^k \leftarrow \pi_h^k(s_h^k)$, receive the reward $r_h(s_h^k, a_h^k)$ and the next state $s_{h+1}^k$
20:      **end for**
21: **end for**
___

Next, we present a lower bound to show the dependency of the total regret on the number of batches for the batch learning model.

**Theorem 4.2.** Suppose that $B \geq (d-1)H/2$. Then for any batch learning algorithm with $B$ batches, there exists a linear MDP such that the regret over the first $T$ rounds is lower bounded by

$$\mathrm{Regret}(T) = \Omega(dH\sqrt{T} + dHT/B).$$

Theorem 4.2 suggests that in order to obtain a standard $\sqrt{T}$-regret, the number of batches $B$ should be at least in the order of $\Omega(\sqrt{T})$, which is similar to its counterpart for batched linear bandits (Han et al., 2020).

## 5 RL in the Rare Policy Switch Model

In this section, we consider the rare policy switch model, where the agent can adaptively choose the batch sizes according to the information collected during the learning process.

### 5.1 Algorithm and Regret Analysis

We first present our second algorithm, LSVI-UCB-RareSwitch, as illustrated in Algorithm 2. Again, due to the nature of linear MDPs, we only need to estimate $\mathbf{w}_h^*$ by ridge regression, and then calculate the optimistic action-value function using the Hoeffding-type exploration bonus $\Gamma_h^k(\cdot, \cdot)$ along with the confidence radius $\beta$. Note that the size of the bonus term in $Q_h^k$ is determined by $\mathbf{\Lambda}_h^k$. Intuitively speaking, the matrix $\mathbf{\Lambda}_h^k$ in Algorithm 2 represents how much information has been learned about the underlying MDP, and the agent only needs to switch the policy after collecting a significant amount of additional information. This is reflected by the determinant of $\mathbf{\Lambda}_h^k$, and the upper confidence bound will become tighter (shrink) as $\det(\mathbf{\Lambda}_h^k)$ increases. The determinant based criterion is similar to the idea of doubling trick, which has been used in the rarely switching OFUL algorithm for stochastic linear bandits (Abbasi-Yadkori et al., 2011), UCRL2 algorithm for tabular MDPs (Jaksch et al., 2010), and UCLK/UCLK+ for linear mixture MDPs in the discounted setting (Zhou et al., 2021b,a).

As shown in Algorithm 2, for each stage $h \in [H]$ the algorithm maintains a matrix $\mathbf{\Lambda}_h$ which is updated at each policy switch (Line 10). For every $k \in [K]$, we denote by $b_k$ the episode from which

the policy $\pi_k$ is computed. This is consistent with the one defined in Algorithm 1 in Section 4. At the start of each episode $k$, the algorithm computes $\{\boldsymbol{\Lambda}_h^k\}_{h\in[H]}$ (Line 5) and then compares them with $\{\boldsymbol{\Lambda}_h\}_{h\in[H]}$ using the determinant-based criterion (Line 7). The agent switches the policy if there exists some $h \in [H]$ such that $\det(\boldsymbol{\Lambda}_h^k)$ has increased by some pre-determined parameter $\eta > 1$, followed by policy evaluation (Lines 11-13). Otherwise, the algorithm retains the previous policy (Line 16). Here the hyperparameter $\eta$ controls the frequency of policy switch, and the total number of policy switches can be bounded by a function of $\eta$.

Algorithm 2 is also a variant of LSVI-UCB proposed in Jin et al. (2020). Compared with LSVI-UCB-Batch in Algorithm 1 for the batch learning model, LSVI-UCB-RareSwitch adaptively decides when to switch the policy and can be tuned by the hyperparameter $\eta$ and therefore fits into the rare policy switch model.

We present the regret bound of Algorithm 2 in the following theorem.

**Theorem 5.1.** There exists some constant $c > 0$ such that for any $\delta \in (0,1)$, if we set $\lambda = 1$, $\beta = cdH\sqrt{\log(2dT/\delta)}$ and $\eta = (1 + K/d)^{dH/B}$, then the number of policy switches $N_{\text{switch}}$ in Algorithm 2 will not exceed $B$. Moreover, the total regret of Algorithm 2 is bounded by

$$
\text{Regret}(T) \leq 2H\sqrt{T\log\left(\frac{2dT}{\delta}\right)} + 2c\sqrt{2d^3H^3T} \cdot \sqrt{\left(\frac{T}{dH}+1\right)^{\frac{dH}{B}}\log\left(\frac{T}{dH}+1\right)\log\left(\frac{2dT}{\delta}\right)}
$$

(5.1)

with probability at least $1 - \delta$.

A few remarks are in order.

**Remark 5.2.** Algorithm 2 needs to update the value of each $\det(\boldsymbol{\Lambda}_h^k)$, and thanks to the special structure of $\boldsymbol{\Lambda}_h^k$, this can be done efficiently by applying the matrix determinant lemma along with the Sherman Morrison formula for efficiently updating each $(\boldsymbol{\Lambda}_h^k)^{-1}$. For simplicity and clarity of the presentation, we do not include these details in the pseudo-code.

**Remark 5.3.** By ignoring the non-dominating term, Theorem 5.1 suggests that the total regret of Algorithm 2 is bounded by $\widetilde{O}(\sqrt{d^3H^3T[1 + T/(dH)]^{dH/B}})$. Also, if we are allowed to choose $B$, we can choose $B = \Omega(dH\log T)$ to achieve $\widetilde{O}(\sqrt{d^3H^3T})$ regret, which is the same as that of LSVI-UCB in Jin et al. (2020). This also significantly improves upon Algorithm 1 when $T$ is sufficiently large since previously we need $B = \Omega(\sqrt{T/dH})$. Our result exhibits a trade-off between the total regret bound and the number of policy switches, i.e., as the adaptivity budget $B$ increases, the regret bound decreases. This will also be reflected by the numerical results later in Section 6.

**Remark 5.4.** Concurrent to our work, Gao et al. (2021) proposed an algorithm with $B = \Omega(dH\log T)$ policy switches. Note that $B = \Omega(dH\log T)$ corresponds to choosing $\eta$ to be a constant, which can be viewed as a special case of our algorithm. Their algorithm does not adapt to different values of budget $B$. Also, they did not study the batch learning model (Section 4) which we think is of equally important practical interest.

**Remark 5.5.** Gao et al. (2021) established a lower bound, which claims that any rare policy switch RL algorithm suffers a linear regret when $B = \widetilde{o}(dH)$. However, unlike our lower bound for the batch learning model (Theorem 4.2), their result does not provide a fine-grained regret lower bound for arbitrary adaptivity constraint $B$. It remains an open problem to establish such kind of lower bound for the rare policy switch model.

## 6 Numerical Experiment

In this section, we provide numerical experiments to support our theory. We run our algorithms, LSVI-UCB-Batch and LSVI-UCB-RareSwitch, on a synthetic linear MDP given in Example 6.1, and compare them with the fully adaptive baseline, LSVI-UCB (Jin et al., 2020).

**Example 6.1** (Hard-to-learn linear MDP, Zhou et al. 2021b)**.** Let $d > 0$ be some integer and $\delta \in (0,1)$ be a constant. The state space $\mathcal{S} = \{0,1\}$ consists of two states, and the action space $\mathcal{A} = \{\pm 1\}^{d-3}$ contains $2^{d-3}$ actions where each action is represented by a $(d-3)$-dimensional

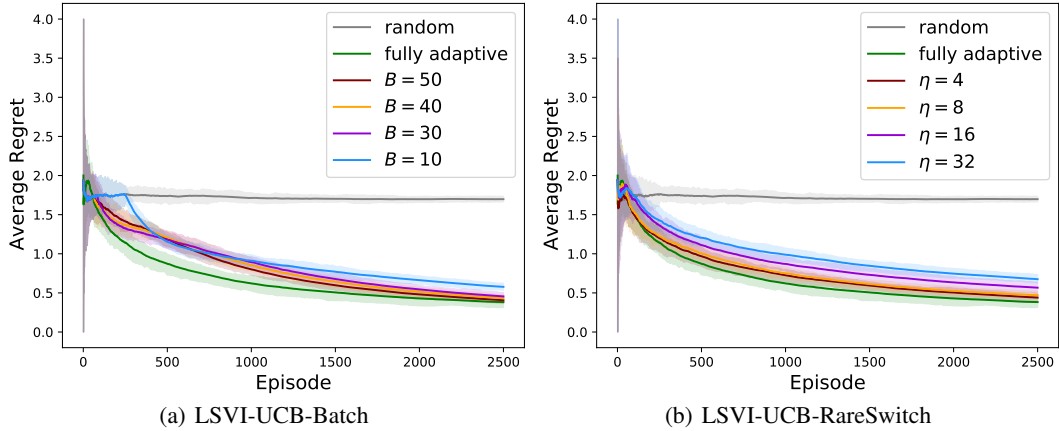

(a) LSVI-UCB-Batch          (b) LSVI-UCB-RareSwitch

Figure 1: Plot of average regret ($\mathrm{Regret}(T)/K$) v.s. the number of episodes. The results are averaged over 50 rounds of each algorithm, and the error bars are chosen to be $[20\%, 80\%]$ empirical confidence intervals.

vector $\mathbf{a}$. For each state-action pair $(s, \mathbf{a}) \in \mathcal{S} \times \mathcal{A}$, the feature vector is given by

$$\phi(s, a) = \begin{cases} (-\mathbf{a}^\top, 1 - \delta, \delta)^\top & s = 0, \\ (0, \dots, 0, \delta, 1 - \delta) & s = 1. \end{cases} \qquad (6.1)$$

For each $h \in [H]$, let $\boldsymbol{\gamma}_h \in \{\pm \delta/(d-2)\}^{d-2}$ and define the corresponding vector-valued measure as

$$\boldsymbol{\mu}_h(s) = \begin{cases} (\boldsymbol{\gamma}_h^\top, 1, 0)^\top & s = 0 \\ (-\boldsymbol{\gamma}_h^\top, 0, 1)^\top & s = 1 \end{cases}. \qquad (6.2)$$

Finally, we set $\boldsymbol{\theta}_h \equiv (0, \dots, 0, -\delta/(1 - 2\delta), (1 - \delta)/(1 - 2\delta)) \in \mathbb{R}^d$ for all $h \in [H]$.

It is straightforward to verify that the feature vectors in (6.1) and the vector-valued measures in (6.2) constitute a valid linear MDP such that, for all $\mathbf{a} \in \mathcal{A}$ and $h \in [H]$,

$$r_h(s, \mathbf{a}) = \mathbb{1}\{s = 1\}, \qquad \mathbb{P}_h(s'|s, \mathbf{a}) = \begin{cases} 1 - \delta - \langle \mathbf{a}, \gamma_h \rangle & (s, s') = (0, 0), \\ \delta + \langle \mathbf{a}, \gamma_h \rangle & (s, s') = (0, 1), \\ \delta & (s, s') = (1, 0), \\ 1 - \delta & (s, s') = (1, 1). \end{cases}$$

In our experiment[6], we set $H = 10$, $K = 2500$, $\delta = 0.35$ and $d = 13$, thus $\mathcal{A}$ contains 1024 actions. Now we apply our algorithms, LSVI-UCB-Batch and LSVI-UCB-RareSwitch, to this linear MDP instance, and compare their performance with the fully adaptive baseline LSVI-UCB (Jin et al., 2020) under different parameter settings. In detail, for LSVI-UCB-Batch, we run the algorithm for $B = 10, 20, 30, 40, 50$ respectively; for LSVI-UCB-RareSwitch, we set $\eta = 2, 4, 8, 16, 32$. We plot the average regret ($\mathrm{Regret}(T)/K$) against the number of episodes in Figure 1. In addition to the regret of the proposed algorithms, we also plot the regret of a uniformly random policy (i.e., choosing actions uniformly randomly in each step) as a baseline.

From Figure 1, we can see that for LSVI-UCB-Batch, when $B \approx \sqrt{K}$, it achieves a similar regret as the fully adaptive LSVI-UCB as it collects more and more trajectories. For LSVI-UCB-RareSwitch, a constant value of $\eta$ yields a similar order of regret compared with LSVI-UCB as suggested by Theorem 5.1. By comparing Figure 1(a) and 1(b), we can see that the performance of LSVI-UCB-RareSwitch is consistently close to that of the fully-adaptive LSVI-UCB throughout the learning process, while the performance gap between LSVI-UCB-Batch and LSVI-UCB is small only when $k$ is large. This suggests a better adaptivity of LSVI-UCB-RareSwitch than LSVI-UCB-Batch, which only updates the policy at prefixed time steps, thus being not adaptive enough.

---

[6]All experiments are performed on a PC with Intel i7-9700K CPU.

Moreover, we can also see the trade-off between the regret and the adaptivity level: with more limited adaptivity (smaller $B$ or larger $\eta$) the regret gap between our algorithms and the fully adaptive LSVI-UCB becomes larger. These results indicate that our algorithms can indeed achieve comparable performance as LSVI-UCB, even under adaptivity constraints. This corroborates our theory.

## 7 Conclusions

In this work, we study online RL with linear function approximation under the adaptivity constraints. We consider both the batch learning model and the rare policy switch models and propose two new algorithms LSVI-UCB-Batch and LSVI-UCB-RareSwitch for each setting. We show that LSVI-UCB-Batch enjoys an $\widetilde{O}(\sqrt{d^3H^3T} + dHT/B)$ regret and LSVI-UCB-RareSwitch enjoys an $\widetilde{O}(\sqrt{d^3H^3T[1 + T/(dH)]^{dH/B}})$ regret. Compared with the fully adaptive LSVI-UCB algorithm (Jin et al., 2020), our algorithms can achieve the same regret with a much fewer number of batches/policy switches. We also prove the regret lower bound for the batch learning learning model, which suggests that the dependency on $B$ in LSVI-UCB-Batch is tight.

For the future work, we would like to prove the regret lower bound for the rare policy switching model that explicitly depends on the given adaptivity budget $B$.

## Acknowledgments and Disclosure of Funding

We would like to thank the anonymous reviewers for their helpful comments. Part of this work was done when DZ and QG participated the Theory of Reinforcement Learning program at the Simons Institute for the Theory of Computing in Fall 2020. DZ and QG are partially supported by the National Science Foundation CAREER Award 1906169, IIS-1904183 and AWS Machine Learning Research Award. The views and conclusions contained in this paper are those of the authors and should not be interpreted as representing any funding agencies.

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
