## A  Additional Details on the Numerical Experiments

### A.1  Log-scaled Plot of the Average Regret

We also provide log-scaled plot of the average regret in Figure 2. We can see that the slope of the average regret curves for our proposed algorithms is similar to that of the fully adaptive LSVI-UCB, all indicating an $\widetilde{O}(1/\sqrt{T})$ scaling.

### A.2  Misspecified Linear MDP

We also empirically evaluate our algorithms on linear MDP with different levels of misspecification. In particular, based on the linear MDP instance constructed in Example 6.1, we follow the definition of $\zeta$-approximate linear MDP in Jin et al. (2020), and consider a corrupted transition given by

$$\mathbb{P}_h(s'|0, a) = (1 - f(a))\boldsymbol{\phi}(0, a)^\top \boldsymbol{\mu}_h(s') + f(a) \mathbb{1}\{s' = g(a)\}$$

where $f : \mathcal{A} \to [0, \zeta]$, $\zeta \in (0, 1)$ and $g : \mathcal{A} \to \mathcal{S}$ are unknown. The two additional functions, $f$ and $g$, can be constructed by random sampling before running the algorithms, and the magnitude of $\zeta \in (0, 1)$ characterizes the level of model misspecification. All the other components of the model and the experiment configurations remain the same as those in Section 6.

Under this misspecified model with levels $\zeta = 0.05, 0.1, 0.2, 0.4$, we run LSVI-UCB-Batch with $B = 50$ and LSVI-UCB-RareSwitch with $\eta = 8$ respectively. We plot the average regret of the algorithms in Figure 3. We can see that our algorithms can still achieve a reasonably good performance under considerable levels of model misspecification.

## B  Proofs of Theorem 4.1

In this section we prove Theorem 4.1

For simplicity, we use $b_k$ to denote the batch $t_b$ satisfying $t_b \leq k < t_{b+1}$. Let $\Gamma_h^k(\cdot, \cdot)$ be $\beta \cdot [\boldsymbol{\phi}(\cdot, \cdot)^\top (\boldsymbol{\Lambda}_h^k)^{-1} \boldsymbol{\phi}(\cdot, \cdot)]^{1/2}$ for any $h \in [H], k \in [K]$. First, we need the following lemma which gives Regret$(T)$ a high probability upper bound that depends on the summation of bonuses.

**Lemma B.1.** With probability at least $1 - \delta$, the total regret of Algorithm 1 satisfies

$$\text{Regret}(T) \leq \sum_{k=1}^{K} \sum_{h=1}^{H} \min \left\{ H, 2\Gamma_h^{b_k}(s_h^k, a_h^k) \right\} + 2H\sqrt{T \log\left(\frac{2dT}{\delta}\right)}.$$

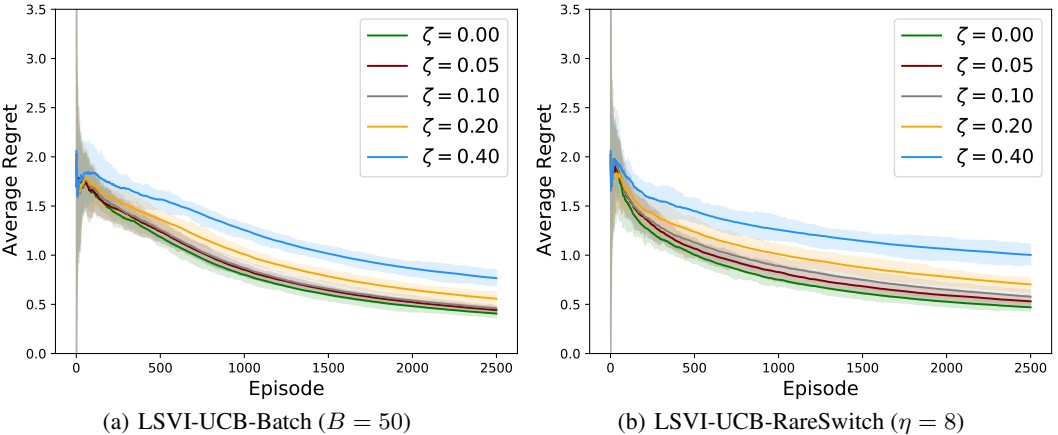

(a) LSVI-UCB-Batch ($B = 50$)        (b) LSVI-UCB-RareSwitch ($\eta = 8$)

Figure 3: Plot of average regret ($\text{Regret}(T)/K$) v.s. the number of episodes for a misspecified linear MDP. The results are averaged over 50 rounds of each algorithm, and the error bars are chosen to be $[20\%, 80\%]$ empirical confidence intervals.

Lemma B.1 suggests that in order to bound the total regret, it suffices to bound the summation of the 'delayed' bonuses $\Gamma_h^{b_k}(s_h^k, a_h^k)$, in contrast to the per-episode bonuses $\Gamma_h^k(s_h^k, a_h^k)$ for all $k \in [K]$. The superscript $b_k$ suggests that instead of using all the information up to the current episode $k$, Algorithm 1 can only use the information before the current batch $b_k$ due to its batch learning nature. How to control the error induced by batch learning is the main difficulty in our analysis. To tackle this difficulty, we first need an upper bound for the summation of per-episode bonuses $\Gamma_h^k(s_h^k, a_h^k)$.

**Lemma B.2.** Let $\beta$ be selected as Theorem 4.1 suggests. Then the summation of all the per-episode bonuses is bounded by

$$\sum_{k=1}^{K} \sum_{h=1}^{H} \Gamma_h^k(s_h^k, a_h^k) \leq \beta \sqrt{2dHT \log\left(\frac{T}{dH} + 1\right)}.$$

It is worth noting that the per-episode bonuses are not generated from our algorithm, but instead are some virtual terms that we introduce to facilitate our analysis. Equipped with Lemma B.2, we only need to bound the difference between delayed bonuses and per-episode bonuses. We consider all the indices $(k, h) \in [K] \times [H]$. The next lemma suggests that considering the ratio between delayed bonuses and per-episode bonuses, the 'bad' indices, where the ratio is large, only appear few times. This is also the key lemma of our analysis.

**Lemma B.3.** Define the set $\mathcal{C}$ as follows

$$\mathcal{C} = \{(k, h) : \Gamma_h^{b_k}(s_h^k, a_h^k)/\Gamma_h^k(s_h^k, a_h^k) > 2\},$$

then we have $|\mathcal{C}| \leq dHK \log(K/d + 1)/(2B \log 2)$.

With all the above lemmas, we now begin to prove our main theorem.

*Proof of Theorem 4.1.* Suppose the event defined in Lemma B.1 holds. Then by Lemma B.1 we have that

$$\text{Regret}(T) \leq \underbrace{\sum_{h=1}^{H} \sum_{k=1}^{K} \min\left\{H, 2\Gamma_h^{b_k}(s_h^k, a_h^k)\right\}}_{I} + 2H\sqrt{T \log\left(\frac{2dT}{\delta}\right)} \qquad \text{(B.1)}$$

holds with probability at least $1 - \delta$. Next, we are going to bound $I$. Let $\mathcal{C}$ be the set defined in Lemma B.3. Then we have

$$
\begin{aligned}
I &= \sum_{(k,h)\in\mathcal{C}} \min\left\{H, 2\Gamma_h^{b_k}(s_h^k, a_h^k)\right\} + \sum_{(k,h)\notin\mathcal{C}} \min\left\{H, 2\Gamma_h^{b_k}(s_h^k, a_h^k)\right\} \\
&\leq H|\mathcal{C}| + 4 \sum_{(k,h)\notin\mathcal{C}} \Gamma_h^k(s_h^k, a_h^k) \\
&\leq H|\mathcal{C}| + 4 \sum_{h=1}^{H}\sum_{k=1}^{K} \Gamma_h^k(s_h^k, a_h^k),
\end{aligned} \tag{B.2}
$$

where the first inequality holds due to the definition of $\mathcal{C}$, and the second one holds trivially. Therefore, substituting (B.2) into (B.1), the regret can be bounded by

$$
\begin{aligned}
\text{Regret}(T) &\leq 2H\sqrt{T\log\left(\frac{2dT}{\delta}\right)} + H|\mathcal{C}| + 4\sum_{h=1}^{H}\sum_{k=1}^{K} \Gamma_h^k(s_h^k, a_h^k) \\
&\leq 2H\sqrt{T\log\left(\frac{2dT}{\delta}\right)} + \frac{dHT}{2B\log 2}\log\left(\frac{T}{dH}+1\right) \\
&\quad + 4c\sqrt{2d^3H^3T\log\left(\frac{2dT}{\delta}\right)\log\left(\frac{T}{dH}+1\right)},
\end{aligned} \tag{B.3}
$$

where the second inequality holds due to Lemmas B.2 and B.3 and the fact that $T = KH$. This completes the proof. $\qquad\square$

## B.1   Proof of Lemma B.1

The following two lemmas in Jin et al. (2020) characterize the quality of the estimates given by the LSVI-UCB-type algorithms.

**Lemma B.4** (Lemma B.5, Jin et al. 2020). *With probability at least $1 - \delta$, we have $Q_h^k(s,a) \geq Q_h^*(s,a)$ for all $(s, a, h, k) \in \mathcal{S} \times \mathcal{A} \times [H] \times [K]$.*

**Lemma B.5** (Lemma B.4, Jin et al. 2020). *There exists some constant $c$ such that if we set $\beta = cdH\sqrt{\log(dT/\delta)}$, then for any fixed policy $\pi$ we have for all $(s, a, h, k) \in \mathcal{S} \times \mathcal{A} \times [H] \times [K]$ that*

$$
\left|\phi(s,a)^\top(\mathbf{w}_h^{b_k} - \mathbf{w}_h^\pi) - \left(\mathbb{P}_h(V_{h+1}^{b_k} - V_{h+1}^\pi)\right)(s,a)\right| \leq \beta\sqrt{\phi(s,a)^\top(\mathbf{\Lambda}_h^{b_k})^{-1}\phi(s,a)}
$$

*with probability at least $1 - \delta$.*

*Proof of Lemma B.1.* By Lemma B.4, we have $Q_h^k(s,a) \geq Q_h^*(s,a)$ for all $(s, a, h, k) \in \mathcal{S} \times \mathcal{A} \times [H] \times [K]$ on some event $\mathcal{E}$ such that $\mathbb{P}(\mathcal{E}) \geq 1 - \delta/2$. In the following argument, all statements would be conditioned on the event $\mathcal{E}$. Then by the definition of $V_1^k$ we know that $V_1^k(s) = \max_{a\in\mathcal{A}} Q_1^k(s,a) \geq \max_{a\in\mathcal{A}} Q_1^*(s,a) = V_1^*(s)$ for all $(s,k) \in \mathcal{S} \times [K]$. Therefore, we have

$$
\text{Regret}(T) = \sum_{k=1}^{K}\left[V_1^*(s_1^k) - V_1^{\pi^k}(s_1^k)\right] \leq \sum_{k=1}^{K}\left[V_1^k(s_1^k) - V_1^{\pi^k}(s_1^k)\right] = \sum_{k=1}^{K}\left[V_1^{b_k}(s_1^k) - V_1^{\pi^k}(s_1^k)\right].
$$

Note that

$$
V_h^{b_k}(s_h^k) - V_h^{\pi^k}(s_h^k) = Q_h^{b_k}(s_h^k, a_h^k) - Q_h^{\pi^k}(s_h^k, a_h^k),
$$

which together with the definition of $Q_h^{b_k}$ and Lemma B.5 implies that

$$
\begin{aligned}
V_h^{b_k}(s_h^k) - V_h^{\pi^k}(s_h^k) &\leq \phi(s_h^k, a_h^k)^\top\mathbf{w}_h^{b_k} - \phi(s_h^k, a_h^k)^\top\mathbf{w}_h^{\pi^k} + \Gamma_h^{b_k}(s_h^k, a_h^k) \\
&\leq \left[\mathbb{P}_h\left(V_{h+1}^{b_k} - V_{h+1}^{\pi^k}\right)\right](s_h^k, a_h^k) + 2\Gamma_h^{b_k}(s_h^k, a_h^k),
\end{aligned}
$$

where the first inequality holds due to the algorithm design, the second one holds due to Lemma B.5. Meanwhile, notice that $0 \leq V_h^{b_k}(s_h^k) - V_h^*(s_h^k) \leq V_h^{b_k}(s_h^k) - V_h^{\pi^k}(s_h^k) \leq H$, then we have

$$
\begin{aligned}
V_h^{b_k}(s_h^k) - V_h^{\pi^k}(s_h^k) &\leq \min\left\{ H, \left[\mathbb{P}_h\left(V_{h+1}^{b_k} - V_{h+1}^{\pi^k}\right)\right](s_h^k, a_h^k) + 2\Gamma_h^{b_k}(s_h^k, a_h^k)\right\} \\
&\leq \left[\mathbb{P}_h\left(V_{h+1}^{b_k} - V_{h+1}^{\pi^k}\right)\right](s_h^k, a_h^k) + \min\left\{H, 2\Gamma_h^{b_k}(s_h^k, a_h^k)\right\} \\
&= V_{h+1}^{b_k}(s_{h+1}^k) - V_{h+1}^{\pi^k}(s_{h+1}^k) + \min\left\{H, 2\Gamma_h^{b_k}(s_h^k, a_h^k)\right\} \\
&\quad + \left[\mathbb{P}_h\left(V_{h+1}^{b_k} - V_{h+1}^{\pi^k}\right)\right](s_h^k, a_h^k) - \left(V_{h+1}^{b_k}(s_{h+1}^k) - V_{h+1}^{\pi^k}(s_{h+1}^k)\right),
\end{aligned}
$$

where the second inequality holds since $V_{h+1}^{b_k} - V_{h+1}^{\pi^k} \geq 0$. Recursively expand the above inequality, and we have

$$
\begin{aligned}
V_1^{b_k}(s_1^k) - V_1^{\pi^k}(s_1^k) &= \sum_{h=1}^{H}\left\{\left[\mathbb{P}_h\left(V_{h+1}^{b_k} - V_{h+1}^{\pi^k}\right)\right](s_h^k, a_h^k) - \left(V_{h+1}^{b_k}(s_{h+1}^k) - V_{h+1}^{\pi^k}(s_{h+1}^k)\right)\right\} \\
&\quad + \sum_{h=1}^{H}\min\left\{H, 2\Gamma_h^{b_k}(s_h^k, a_h^k)\right\}.
\end{aligned}
$$

Therefore, the total regret can be bounded as follows

$$
\begin{aligned}
\text{Regret}(T) &\leq \sum_{k=1}^{K}\sum_{h=1}^{H}\left\{\left[\mathbb{P}_h\left(V_{h+1}^{b_k} - V_{h+1}^{\pi^k}\right)\right](s_h^k, a_h^k) - \left(V_{h+1}^{b_k} - V_{h+1}^{\pi^k}\right)(s_{h+1}^k)\right\} \\
&\quad + \sum_{k=1}^{K}\sum_{h=1}^{H}\min\left\{H, 2\Gamma_h^{b_k}(s_h^k, a_h^k)\right\}.
\end{aligned}
$$

Note that conditional on $\mathcal{F}_{k,h,1}$, $V_{h+1}^{b_k}$ and $V_{h+1}^{\pi^k}$ are both deterministic, while $s_{h+1}^k$ follows the distribution $\mathbb{P}_h(\cdot|s_h^k, a_h^k)$. Therefore, the first term on the RHS is a sum of a martingale difference sequence such that each summand has absolute value at most $2H$. Applying Azuma-Hoeffding inequaliy yields

$$
\sum_{k=1}^{K}\sum_{h=1}^{H}\left\{\left[\mathbb{P}_h\left(V_{h+1}^{b_k} - V_{h+1}^{\pi^k}\right)\right](s_h^k, a_h^k) - \left(V_{h+1}^{b_k} - V_{h+1}^{\pi^k}\right)(s_{h+1}^k)\right\} \leq 2H\sqrt{T\log\left(\frac{2dT}{\delta}\right)},
$$

with probability at least $1 - \delta/2$. By a union bound over the event $\mathcal{E}$ and the convergence of the martingale, with probability at least $1 - \delta$, we have

$$
\text{Regret}(T) \leq 2H\sqrt{T\log\left(\frac{2dT}{\delta}\right)} + \sum_{k=1}^{K}\sum_{h=1}^{H}\min\left\{H, 2\Gamma_h^{b_k}(s_h^k, a_h^k)\right\}.
$$

$\square$

## B.2 Proof of Lemma B.2

We need the following lemma to bound the sum of the bonus terms.

**Lemma B.6** (Lemma 11, Abbasi-Yadkori et al. 2011)**.** Let $\{\phi_t\}_{t=1}^{\infty}$ be an $\mathbb{R}^d-$valued sequence. Meanwhile, let $\Lambda_0 \in \mathbb{R}^{d \times d}$ be a positive-definite matrix and $\Lambda_t = \Lambda_0 + \sum_{i=1}^{t-1}\phi_i\phi_i^\top$. It holds for any $t \in \mathbb{Z}_+$ that

$$
\sum_{i=1}^{t}\min\{1, \phi_i^\top\Lambda_i^{-1}\phi_i\} \leq 2\log\left(\frac{\det(\Lambda_{t+1})}{\det(\Lambda_1)}\right).
$$

Moreover, assuming that $\|\phi_i\|_2 \leq 1$ for all $i \in \mathbb{Z}_+$ and $\lambda_{\min}(\Lambda_0) \geq 1$, it holds for any $t \in \mathbb{Z}_+$ that

$$
\log\left(\frac{\det(\Lambda_{t+1})}{\det(\Lambda_1)}\right) \leq \sum_{i=1}^{t}\phi_i^\top\Lambda_i^{-1}\phi_i \leq 2\log\left(\frac{\det(\Lambda_{t+1})}{\det(\Lambda_1)}\right).
$$

*Proof of Lemma B.2.* We can bound the summation of $\Gamma_h^k(s_h^k, a_h^k)$ as follows:

$$\sum_{h=1}^{H}\sum_{k=1}^{K}\Gamma_h^k(s_h^k, a_h^k) \leq \sum_{h=1}^{H}\sqrt{K\cdot\sum_{k=1}^{K}[\Gamma_h^k(s_h^k, a_h^k)]^2} = \beta\sqrt{K}\sum_{h=1}^{H}\sqrt{\sum_{k=1}^{K}\phi(s_h^k, a_h^k)^\top[\mathbf{\Lambda}_h^k]^{-1}\phi(s_h^k, a_h^k)},$$

where the inequality holds due to Cauchy-Schwarz inequality. Furthermore, by Lemma B.6, we have

$$\sum_{k=1}^{K}\phi(s_h^k, a_h^k)^\top[\mathbf{\Lambda}_h^k]^{-1}\phi(s_h^k, a_h^k) \leq 2\log\left(\frac{\det\mathbf{\Lambda}_h^{K+1}}{\det\mathbf{\Lambda}_h^1}\right) \leq 2d\log(K/d+1),$$

where the second inequality holds due to Lemma C.1. That finishes our proof. $\qquad\square$

### B.3 Proof of Lemma B.3

*Proof of Lemma B.3.* First, let $\mathcal{C}_h$ denote the indices $k$ where $(k, h) \in \mathcal{C}$, then we have $|\mathcal{C}| = \sum_{h=1}^{H}|\mathcal{C}_h|$. Next we bound $|\mathcal{C}_h|$ for each $h$. For each $k \in \mathcal{C}_h$, suppose $t_b \leq k < t_{b+1}$, then we have $b_k = t_b$ and

$$\log\det(\mathbf{\Lambda}_h^{t_{b+1}}) - \log\det(\mathbf{\Lambda}_h^{t_b}) \geq \log\det(\mathbf{\Lambda}_h^k) - \log\det(\mathbf{\Lambda}_h^{b_k}) \geq 2\log(\Gamma_h^{b_k}(s_h^k, a_h^k)/\Gamma_h^k(s_h^k, a_h^k)) > 2\log 2,$$

where the first inequality holds since $\mathbf{\Lambda}_h^{t_{b+1}} \succeq \mathbf{\Lambda}_h^k$, the second inequality holds due to Lemma C.2, the third one holds due to the definition of $\mathcal{C}_h$. Thus, let $\widehat{\mathcal{C}}_h$ denote the set

$$\widehat{\mathcal{C}}_h = \{b \in [B] : \log\det(\mathbf{\Lambda}_h^{t_{b+1}}) - \log\det(\mathbf{\Lambda}_h^{t_b}) > 2\log 2\},$$

we have $|\mathcal{C}_h| \leq \lfloor K/B\rfloor \cdot |\widehat{\mathcal{C}}_h|$. In the following we bound $|\widehat{\mathcal{C}}_h|$. Now we consider the sequence $\{\log\det(\mathbf{\Lambda}_h^{t_{b+1}}) - \log\det(\mathbf{\Lambda}_h^{t_b})\}$. It is easy to see $\log\det(\mathbf{\Lambda}_h^{t_{b+1}}) - \log\det(\mathbf{\Lambda}_h^{t_b}) \geq 0$, therefore

$$2\log 2|\widehat{\mathcal{C}}_h| \leq \sum_{b\in\widehat{\mathcal{C}}_h}[\log\det(\mathbf{\Lambda}_h^{t_{b+1}}) - \log\det(\mathbf{\Lambda}_h^{t_b})] \leq \sum_{b=1}^{B}[\log\det(\mathbf{\Lambda}_h^{t_{b+1}}) - \log\det(\mathbf{\Lambda}_h^{t_b})]. \quad \text{(B.4)}$$

Meanwhile, we have

$$\sum_{b=1}^{B}[\log\det(\mathbf{\Lambda}_h^{t_{b+1}}) - \log\det(\mathbf{\Lambda}_h^{t_b})] = \log\det(\mathbf{\Lambda}_h^{t_{B+1}}) = \log\det(\mathbf{\Lambda}_h^{K+1}) \leq d\log(K/d+1),$$

(B.5)

where the last inequality holds due to Lemma C.1. Therefore, (B.4) and (B.5) suggest that $|\widehat{\mathcal{C}}_h| \leq d\log(K/d+1)/(2\log 2)$. Finally, we bound $|\mathcal{C}|$ as follows, which ends our proof.

$$|\mathcal{C}| = \sum_{h=1}^{H}|\mathcal{C}_h| \leq \sum_{h=1}^{H}K/B\cdot|\widehat{\mathcal{C}}_h| \leq dHK\log(K/d+1)/(2B\log 2).$$

$\qquad\square$

## C Proof of Theorem 5.1

Now we provide the proof of Theorem 5.1. We continue to use the notions that have been introduced in Section 4. We first give an upper bound on the determinant of $\mathbf{\Lambda}_h^k$.

**Lemma C.1.** Let $\{\mathbf{\Lambda}_h^k, (k, h) \in [K] \times [H]\}$ be as defined in Algorithms 1 and 2. Then for all $h \in [H]$ and $k \in [K]$, we have $\det(\mathbf{\Lambda}_h^k) \leq (\lambda + (k-1)/d)^d$.

*Proof.* Note that

$$\text{tr}(\mathbf{\Lambda}_h^k) = \text{tr}(\lambda\mathbf{I}_d) + \sum_{\tau=1}^{k-1}\text{tr}\left(\phi(s_h^\tau, a_h^\tau)\phi(s_h^\tau, a_h^\tau)^\top\right) = \lambda d + \sum_{\tau=1}^{k-1}\|\phi(s_h^\tau, a_h^\tau)\|_2^2 \leq \lambda d + k - 1,$$

where the inequality follows from the assumption that $\|\phi(s,a)\|_2 \le 1$ for all $(s,a) \in \mathcal{S} \times \mathcal{A}$. Since $\mathbf{\Lambda}_h^k$ is positive semi-definite, by inequality of arithmetic and geometric means, we have

$$\det(\mathbf{\Lambda}_h^k) \le \left(\frac{\operatorname{tr}(\mathbf{\Lambda}_h^k)}{d}\right)^d \le \left(\lambda + \frac{k-1}{d}\right)^d.$$

This finishes the proof. □

Next lemma provides a determinant-based upper bound for the ratio between the norms $\|\cdot\|_{\mathbf{A}}$ and $\|\cdot\|_{\mathbf{B}}$, where $\mathbf{A} \succeq \mathbf{B}$.

**Lemma C.2** (Lemma 12, Abbasi-Yadkori et al. 2011)**.** Suppose $\mathbf{A}, \mathbf{B} \in \mathbb{R}^{d \times d}$ are two positive definite matrices satisfying that $\mathbf{A} \succeq \mathbf{B}$, then for any $\mathbf{x} \in \mathbb{R}^d$, we have $\|\mathbf{x}\|_{\mathbf{A}} \le \|\mathbf{x}\|_{\mathbf{B}} \cdot \sqrt{\det(\mathbf{A})/\det(\mathbf{B})}$.

The switching cost of Algorithm 2 is characterized in the following lemma.

**Lemma C.3.** For any $\eta > 1$ and $\lambda > 0$, the global switching cost of Algorithm 2 is bounded by

$$N_{\text{switch}} \le \frac{dH}{\log \eta} \log\left(1 + \frac{K}{\lambda d}\right).$$

*Proof.* Let $\{k_1, k_2, \cdots, k_{N_{\text{switch}}}\}$ be the episodes where the algorithm updates the policy, and we also define $k_0 = 0$. Then by the determinant-based criterion (Line 7), for each $i \in [N_{\text{switch}}]$ there exists at least one $h \in [H]$ such that

$$\det(\mathbf{\Lambda}_h^{k_i}) > \eta \cdot \det(\mathbf{\Lambda}_h^{k_{i-1}}).$$

By the definition of $\mathbf{\Lambda}_h^k$ (Line 5), we know that $\mathbf{\Lambda}_h^{j_1} \succeq \mathbf{\Lambda}_h^{j_2}$ for all $j_1 \ge j_2$ and $h \in [H]$. Thus we further have

$$\prod_{h=1}^H \det(\mathbf{\Lambda}_h^{k_i}) > \eta \cdot \prod_{h=1}^H \det(\mathbf{\Lambda}_h^{k_{i-1}}).$$

Applying the above inequality for all $i \in [N_{\text{switch}}]$ yields

$$\prod_{h=1}^H \det\left(\mathbf{\Lambda}_h^{k_{N_{\text{switch}}}}\right) > \eta^{N_{\text{switch}}} \cdot \prod_{h=1}^H \det(\mathbf{\Lambda}_h^0) = \eta^{N_{\text{switch}}} \lambda^{dH},$$

as we initialize $\mathbf{\Lambda}_h^0$ to be $\lambda \mathbf{I}_d$. While by Lemma C.1, we have

$$\prod_{h=1}^H \det\left(\mathbf{\Lambda}_h^{k_{N_{\text{switch}}}}\right) \le \prod_{h=1}^H \det(\mathbf{\Lambda}_h^K) \le \left(\lambda + \frac{K}{d}\right)^{dH}.$$

Therefore, combining the above two inequalities, we obtain that

$$N_{\text{switch}} \le \frac{dH}{\log \eta} \log\left(1 + \frac{K}{\lambda d}\right).$$

This completes the proof. □

We now begin to prove our main theorem.

*Proof of Theorem 5.1.* First, substituting the choice of $\eta$ and $\lambda = 1$ into the bound in Lemma C.3 yields that $N_{\text{switch}} \le B$.

Next, we bound the regret of Algorithm 2. The result of Lemma B.1 still holds here, thus it suffices to bound the summation of the bonus terms $\Gamma_h^{b_k}(s_h^k, a_h^k)$. Note that $b_k \le k$, and thus $\mathbf{\Lambda}_h^k \succeq \mathbf{\Lambda}_h^{b_k}$ for all $(h,k) \in [H] \times [K]$. Then by Lemma C.2 we have

$$\frac{\Gamma_h^{b_k}(s_h^k, a_h^k)}{\Gamma_h^k(s_h^k, a_h^k)} \le \sqrt{\frac{\det(\mathbf{\Lambda}_h^k)}{\det(\mathbf{\Lambda}_h^{b_k})}} \le \sqrt{\eta} \qquad (\text{C.1})$$

for all $(h, k) \in [H] \times [K]$, where the second inequality holds due to the algorithm design. Hence, we have

$$\sum_{k=1}^{K}\sum_{h=1}^{H}\Gamma_h^{b_k}(s_h^k, a_h^k) \leq \sqrt{\eta}\sum_{k=1}^{K}\sum_{h=1}^{H}\Gamma_h^k(s_h^k, a_h^k) \leq \beta\sqrt{2\eta dHT\log\left(\frac{T}{dH}+1\right)},$$

where the second inequality follows from Lemma B.2. Therefore, we conclude by Lemma B.1 that

$$\text{Regret}(T) \leq 2c\sqrt{2\eta d^3 H^3 T\log\left(\frac{T}{dH}+1\right)\log\left(\frac{2dT}{\delta}\right)} + 2H\sqrt{T\log\left(\frac{2dT}{\delta}\right)} \qquad \text{(C.2)}$$

holds with probability at least $1 - \delta$. Finally, substituting the choice of $\eta$ into (C.2) finishes our proof. $\qquad\square$

## D  Proofs of Theorem 4.2

In this section, we prove the lower bound for the batch learning model.

*Proof of Theorem 4.2.* We prove the $\Omega(dH\sqrt{T})$ and $\Omega(dHT/B)$ lower bounds separately. The first term has been proved in Theorem 5.6, (Zhou et al., 2021a). In the remaining of this proof, we prove the second term. We consider a class of MDPs parameterized by $\gamma \in \Gamma \subset \mathbb{R}^{2dH}$, where $\Gamma$ is defined as follows

$$\Gamma = \left\{(\mathbf{b}_{1,1}^\top, \cdots, \mathbf{b}_{H,d}^\top)^\top : \mathbf{b}_{i,j} \in \{(0,1)^\top, (1,0)^\top\}\right\}.$$

The MDP is defined as follows. The states space $\mathcal{S}$ consist of has $d + 1$ states $x_0, \cdots, x_d$, and the action space $\mathcal{A}$ contains two actions $\mathbf{a}_1 = (0,1)^\top, \mathbf{a}_2 = (1,0)^\top$. For any $\gamma = (\mathbf{b}_{1,1}^\top, \cdots, \mathbf{b}_{H,d}^\top)^\top$, the feature mapping is defined as

$$\phi(x_0, \mathbf{a}_j) = (1, \underbrace{0, \cdots, 0}_{2d})^\top, \qquad \phi(x_i, \mathbf{a}_j) = (1, \underbrace{0, \cdots, 0}_{2i-2}, \mathbf{a}_j^\top, \underbrace{0, \cdots, 0}_{2d-2i})^\top \in \mathbb{R}^{2d+1}$$

for every $i \in [d]$ and $j \in \{1, 2\}$. We further define the vector-valued measures as

$$\boldsymbol{\mu}_h^\gamma(x_0) = (1, -\mathbf{b}_{h,1}^\top, \cdots, -\mathbf{b}_{h,d}^\top)^\top, \qquad \boldsymbol{\mu}_h^\gamma(x_i) = (\underbrace{0, \cdots, 0}_{2i-1}, \mathbf{b}_{h,i}^\top, \underbrace{0, \cdots, 0}_{2d-2i})^\top$$

for every $i \in [d]$, $j \in \{1, 2\}$ and $h \in [H]$. Finally, for each $h \in [H]$, we define

$$\boldsymbol{\theta}_h = (0, \underbrace{1, \cdots, 1}_{2d})^\top \in \mathbb{R}^{2d+1}.$$

Thereby, for each $h \in [H]$, the transition $\mathbb{P}_h^\gamma$ is defined as $\mathbb{P}_h^\gamma(s'|s, \mathbf{a}) = \langle \phi(s, \mathbf{a}), \boldsymbol{\mu}_h^\gamma(s') \rangle$, and the reward function is $r_h(s, \mathbf{a}) = \langle \phi(s, \mathbf{a}), \boldsymbol{\theta}_h \rangle$ for all $(s, \mathbf{a}) \in \mathcal{S} \times \mathcal{A}$. It is straightforward to see that the reward satisfies $r_h(x_0, \mathbf{a}) = 0$ and $r_h(x_i, \mathbf{a}) = 1$ for $i \in [d]$ and all $\mathbf{a} \in \mathcal{A}$. In addition, the starting state can be $x_0$ or $x_i$.

Based on the above definition, we have the following transition dynamic:

- $x_0$ is an absorbing state.

- For any $i \in [d]$, $x_i$ can only transit to $x_0$ or $x_i$.

- For any episode starting from $x_0$, there is no regret.

- For any episode starting from some $x_i$ with $i \in [d]$, suppose $h$ is the first stage where the agent did not choose the "right" action $\mathbf{a} = \mathbf{b}_{h,i}$, then the regret for this episode is $H - h$.

Now we show that for any deterministic algorithm[7], there exists a $\gamma \in \Gamma$ such that the regret is lower bounded by $dHT/B$. Suppose $1 = t_1 < \cdots < t_{B+1} = K + 1$. We can treat all episodes in the same

---

[7]The lower bound of random algorithms is lower bounded by the lower bound of deterministic algorithms according to Yao's minimax principle.

batch as copies of one episode, because all actions taken by the agent, transitions and rewards are the same. When $B \geq dH$, there exists $\mathcal{C} = \{c_{1,1}, \cdots, c_{H,d}\} \subset [B]$ with $|\mathcal{C}| = dH$ such that

$$\sum_{h \in [H]} \sum_{j \in [d]} (t_{c_{h,j}+1} - t_{c_{h,j}}) \geq \frac{dHK}{B}.$$

For simplicity, we denote the $i$-th batch as the collection of episodes $\{t_i, \cdots, t_{i+1} - 1\}$. Now we carefully pick the starting state $s_0^i$ for the episodes in the $i$-th batch.

- For any batch whose starting episode does not belong to $\mathcal{C}$, we set the starting states of the episodes in this batch as $x_0$. In other words, for $i \notin \mathcal{C}$, we set $s_0^{t_i} = \cdots = s_0^{t_{i+1}-1} = x_0$.

- For any batch whose starting episode lies in $\mathcal{C}$, for $i = c_{h,j} \in \mathcal{C}$, we set $s_0^{t_{c_{h,j}}} = \cdots = s_0^{t_{c_{h,j}+1}-1} = x_j$.

We consider the regret over batches $c_{1,i}, \cdots, c_{H,i}$. Since the algorithm, transition and reward are all deterministic, then the environment can predict the agent's selection. Specifically, suppose the agent will always take action $\mathbf{a}$ at $h$-th stage in the episodes belonging to the $c_{h,j}$-th batch, where $h \leq H/2$. Then the environment selects $\mathbf{b}_{h,j}$ as $(1,1)^\top - \mathbf{a}$, i.e., the other action. Therefore, the agent will always pick the "wrong" action when she firstly visits state $x_j$ at $h$-th stage, which occurs at least $H - h \geq H/2$ regret. Moreover, since for the batch learning model, all the actions are decided at the beginning of each batch, then the $H/2$ regret will last $(t_{c_{h,j}+1} - t_{c_{h,j}})$ episodes. Taking the summation, we have

$$\text{Regret}(T) \geq \frac{H}{2} \cdot \sum_{h \in [H]} \sum_{j \in [d]} (t_{c_{h,j}+1} - t_{c_{h,j}}) \geq \frac{dHT}{2B}.$$

Finally, replacing $d$ by $(d-1)/2$, we can convert our feature mapping from a $(2d+1)$-dimensional vector to a $d$-dimensional vector and complete the proof. $\qed$