# OpenReview forum: "Provably Efficient Reinforcement Learning with Linear Function Approximation under Adaptivity Constraints"
_NeurIPS.cc/2021/Conference — NeurIPS 2021 Poster_

### Official Review · Reviewer_FRs1 · 2021-07-12

**Rating:** 6
**Confidence:** 4

**Summary:**

This paper studies linear mdp under two adaptivity models: rare policy switch model and batch learning model. They achieve the same regret as the fully adaptivity setting (LSVI-UCB) with much less number of policy switches ( e.g., in the rare policy switch model, it is $dH \log T$ vs $T$)

**Ethical Concerns:**

No concern.

**Limitations And Societal Impact:**

The authors have adequately addressed the limitations and social impact of their work.



**Main Review:**

**Strong points**
-	A new result of a sample efficient linear RL with limited adaptivity that seems more comprehensive (as it studies batch learning model as well) than the concurrent work Gao et al. 2021. It also encompasses a result in Gao et al. 2021.
- This is a complete study with two proposed algorithms and corresponding regret analysis, and lower bound analysis.

**Weak points**
-	The approach and analysis are quite simple as they adopt from Abbasi-Yadkori et al. 2021, Jin et al. 2020 that I am not sure if there is any new technical idea.  For example, Algorithm 2 in the present paper is the same as the algorithm in Figure 3 in Abbasi-Yadkori et al. 2021 except that the former is for linear MDP and the latter is for linear bandits.

**Minor comments**
-	Fig 1: the colours for B=50 and B=15 are difficult to distinguish. Moreover, it would be nice to have a baseline line there to give the readers an idea of how much regret is “bad”.  The current experimental result shows that LSVI-UCB-Batch and LSVI-UCB-RareSwtich underperform their full adaptivity counterpart LSVI-UCB. Without a baseline, it could be difficult to know if such a gap is substaintial. Moreover, line 312, the authors say that LSVI-UCB-Batch achieves the similar regret as LSVI-UCB, but Fig 1.(a) does not indicate so.

- The current experiment only shows two configurations (B=15,50 and eta=4,8) for each proposed algorithm. It would be an addition if the results for more configurations are presented to show the trade-offs between regret and adaptivity level.

**Questions**:
-	Line 88-91: “Our goal is to design efficient RL algorithms under a switching cost budge B while their goal is to achieve the optimal rate in terms of T”: How are these two goals different?
-	I would like the authors to comment on the difference between Algorithm 2 of this paper and Algorithm 1 of Gao et al. 2021, and compare their advantages/efficiency in solving the adaptivity constraint problem.



**Time Spent Reviewing:**

5

---

> ### Author Response · Authors · 2021-08-10
> **Response to Reviewer FRs1**
>
> **Q1**: ‘Fig 1: the colours for B=50 and B=15 are difficult to distinguish.’
>
> **A1**: We will adjust the color for better visualization.
>
> ---
>
> **Q2**: ‘Moreover, line 312, the authors say that LSVI-UCB-Batch achieves the similar regret as LSVI-UCB, but Fig 1.(a) does not indicate so.’
>
> **A2**: We are sorry for the statement being unclear and misleading. Indeed, when $B$ is small, the difference in regret between LSVI-UCB-Batch and LSVI-UCB is substantial. Yet, with a larger $B$, the regret of LSVI-UCB-Batch for larger $T$’s is similar to that of LSVI-UCB. We will clarify the statement in the revision.
>
> ---
>
> **Q3**: ‘Moreover, it would be nice to have a baseline line there to give the readers an idea of how much regret is “bad”.’ ‘It would be an addition if the results for more configurations are presented to show the trade-offs between regret and adaptivity level.’
>
> **A3**: Thank you for the suggestion! In the revision, we will add a baseline of the pure random policy (i.e. choosing actions uniformly randomly) and a few more levels of $B$ and $\eta$ in the plots. In fact, in our experiments, the average regret of the uniformly random policy is close to 2, which is substantially larger than those of the algorithms that we have already plotted in the figure.
>
> ---
>
> **Q4**: ‘Line 88-91: “Our goal is to design efficient RL algorithms under a switching cost budge B while their goal is to achieve the optimal rate in terms of T”: How are these two goals different?’
>
> **A4**: In our algorithm, we have a hyperparameter $B$ that is the number of policy switches, while [Gao et al., 2021] does not have such flexibility to choose the number of policy switches and their goal was to achieve sublinear regret with as few policy switches as possible. Note that in practical applications (e.g., clinical trial), the number of policy switches is often designed by the clinician, so we think it is important to treat $B$ as a hyperparameter.
>
> ---
>
> **Q5**: ‘I would like the authors to comment on the difference between Algorithm 2 of this paper and Algorithm 1 of Gao et al. 2021, and compare their advantages/efficiency in solving the adaptivity constraint problem.’
>
> **A5**: First, as we mentioned above, there is no budget parameter $B$ in Algorithm 1 of [Gao et al., 2021], thus their algorithm would fail when such a $B$ is mandatory. On the contrary, our Algorithm 2 LSVI-UCB-RareSwitch takes $B$ as a hyperparameter and thus can adapt to different adaptivity levels. Their Algorithm 1 is exactly a special case of our LSVI-UCB-RareSwitch by setting $\eta$ to be roughly a constant. Moreover, they did not study the batch learning model which we think is also of equally important practical interest. In terms of computational efficiency, their Algorithm 1 and our Algorithm 2 have the same computational complexity.

---

> > ### Comment · Reviewer_FRs1 · 2021-08-24
> > **Response to the author's comment**
> >
> > I read the author's response and other reviews. I am satisfied with the author's response and will keep my initial score. I encourage the authors to take the comments to the paper.

---

> > > ### Author Response · Authors · 2021-08-25
> > > **Thank you for your positive feedback!**
> > >
> > > We will be sure to revise our paper according to your and other reviewer's comments in the final version.

---

### Official Review · Reviewer_3Uwm · 2021-07-16

**Rating:** 6
**Confidence:** 4

**Summary:**

This paper considers sample-efficient RL in linear MDPs under adaptivity constraints, that is, the number of policy switches during the algorithm execution is constrained to be low. The paper considers two types of adaptivity constraint: the batch learning model (in which the switching schedule is predetermined), and the rare policy switch model (in which the switching schedule can be adaptive). The main contributions of this paper are low-regret algorithms for linear MDPs in both versions of low switching models.

**Limitations And Societal Impact:**

Yes.

**Main Review:**

This paper is an extension of the previous works on low switching cost learning in bandits and tabular MDPs. I think overall the results make good contributions to the literature and contain some interesting theoretical insights. The algorithms for both switching models are fairly clean (the Batch learning model just uses a delayed Q update; the Rare Policy Switch model uses an adaptive switching criterion that depends on the determinant of the cumulative feature covariance matrix). These are analogous to (and not exactly the same as) the per-state exponential switching schedule of Bai et al. 2019 for tabular MDPs.

Furthermore the paper has a small numerical experiment section, which I do appreciate as for an RL theory paper. The particular experiments here show that the low switching cost algorithms do achieve sublinear regret. I wonder if the authors could check whether the avg regret scales as $O(1/\sqrt{T})$? (e.g. by looking at slope on log scale)?

My only concern is that the results sound a bit incremental --- I wonder in what sense are the techniques new and different from existing analysis on low switching cost? E.g. by looking at Lemmas A.1, A.2 I get the feeling that the consequence of delayed update here in linear MDP is easier to analyze than in tabular cases, because of the global switching schedule (so that there is only $b_k$, not something like $b_k(s, a)$). Maybe Lemma A.3 and Lemma B.3 may be the key non-trivial ones that allow the result? Could the authors discuss some of the intuitions about these Lemmas and in particular which parts are new? For example, how Lemma B.3 compares with the per-state exponential scheduling in existing low switching cost algorithms for tabular MDPs.


**Time Spent Reviewing:**

2

---

> ### Author Response · Authors · 2021-08-10
> **Response to Reviewer 3Uwm**
>
> **Q1**: ‘I wonder if the authors could check whether the avg regret scales as $\tilde O(1/\sqrt{T})$?’
>
> **A1**: Yes, if we plot the average regret in log scale, we can see that it approximately scales as $\tilde O(1/\sqrt{T})$. We will add this plot in the revision.
>
> ---
>
> **Q2**: ‘My only concern is that the results sound a bit incremental --- I wonder in what sense are the techniques new and different from existing analysis on low switching cost?...because of the global switching schedule.’
>
> **A2**: Compared with the existing works on low switching cost in RL, the adaptivity constraint setting we consider here is more general, which is characterized by the number of policy switches $B$. The low switching cost setting considered in [Bai et al. 2019] and [Gao et al. 2021] can be seen as a special case of ours when $B=\log(T)$. In addition, since we study more general linear MDPs rather than tabular MDPs, where the exploration bonus is constructed as $\sqrt{\phi(s,a)^\top \Lambda_h^k \phi(s,a)}$, naturally we need to control the inflation of these bonus terms when the update is delayed. Moreover, [Bai et al. 2019] and [Gao et al. 2020] did not consider the uniform batch grids and our work is the first to analyze the performance of RL algorithms in this setting.
>
> On the other hand, another way to comprehend the difference between our work and [Bai et al. 2019] is by using one-hot feature mappings to specialize the linear MDP to the tabular MDP (Example 2.1 in [Jin et al. 2020]). Then it is easy to verify that $\Lambda_h^k$ is a diagonal matrix of which each diagonal entry is the visit count of each state-action pair. In this regard, the switching criterion adopted in [Bai et al. 2019] is also a special case of the more general switching criterion used in LSVI-UCB-RareSwitch.
>
> ---
>
> **Q3**: ‘Maybe Lemma A.3 and Lemma B.3 may be the key non-trivial ones that allow the result? Could the authors discuss some of the intuitions about these Lemmas and in particular which parts are new?’
>
> **A3**: Yes, Lemma A.3 and Lemma B.3 are the key lemmas in the proofs of the corresponding regret upper bounds. The intuition behind these two lemmas is that the amount of information obtained through the interaction between the agent and the environment is diminishing, i.e., the more the agent interacts with the environment, the less information the agent gains during the same period of time. (This is kind of related to the phenomenon called ‘diminishing marginal utility’ in statistics and information theory.) Then if we translate this observation into the algorithmic design for linear MDPs, it is reflected by the covariance matrix $\Lambda_h^k$ of which the change would be gradual and less significant as $k$ grows. As we have pointed out before, the per-$(s,a)$ exponential scheduling used in [Bai et al. 2019] is a special case of our determinant-based criterion in LSVI-UCB-RareSwitch. Also, our analysis is succinct yet generic, and we believe that such ideas can be applied to more general RL settings such as general function approximation [Wang et al. 2020], as well as other kinds of MDPs (e.g., linear mixture MDPs [Ayoub et al. 2020]).

---

> > ### Comment · Reviewer_3Uwm · 2021-08-30
> > **Response to authors**
> >
> > Thank you for the response and the explanations about the technical differences. Overall I am on the positive side about this paper and will keep my score.

---

> > > ### Author Response · Authors · 2021-09-02
> > > **Thank you for your positive feedback!**
> > >
> > > We will revise the writing and add more clarifications in the final version according to your comments.

---

### Official Review · Reviewer_ZCnH · 2021-07-17

**Rating:** 6
**Confidence:** 3

**Summary:**

This paper proposed two algorithms for reinforcement learning with linear function approximation under adaptivity constraints.

**Limitations And Societal Impact:**

This is a theoretical paper, and I think there is no potential negative social impact.

**Main Review:**

Strengths:
This papers proves upper bounds on the regret in both the batch learning model and rare policy switch model, which illustrates a trade-off between the regret and global switching cost. It also shows a lower bound in the batch learning model.

Weakness / comments:
1. It seems that the upper bound in [1] is a special case of Theorem 5.1 when the dependency of regret on $T$ is $\sqrt{T}$. But the paper does not explain whether this difference of theoretical results comes from the design of the algorithm or the proof techniques. A clearer and complete comparison between [1] and the current paper would be helpful.

2. The lower bound in Theorem 4.2 assumes $B\geq(d-1)H/2$, while in order to make the upper bound in Theorem 4.1 on the order of $\widetilde{O}(\sqrt{d^3H^3T})$, it is required that $B=\Omega(\sqrt{T/dH})$. Does this mean the upper bound and lower bound only match when $T=O(d^3H^3)$? If it does, how restrictive is this condition?

[1] Gao, Minbo, et al. "A provably efficient algorithm for linear markov decision process with low switching cost." arXiv preprint arXiv:2101.00494 (2021).

**Time Spent Reviewing:**

1.5

---

> ### Author Response · Authors · 2021-08-10
> **Response to Reviewer ZCnH**
>
> **Q1**: ‘A clearer and complete comparison between [Gao et al., 2021] and the current paper would be helpful.’
>
> **A1**: To compare our work with [Gao et al., 2021], we would like to remark that their main algorithm is a special case of our LSVI-UCB-RareSwitch by setting $\eta$ to be roughly a constant. Their proposed algorithm does not adapt to different levels of budget $B$, while ours can be determined by the user. Our analysis is more general yet succinct. Moreover, they did not study the batching learning model which we think is also of important practical interest. We will add more detailed comments in the revision as suggested.
>
> ---
>
> **Q2**: ‘Does this mean the upper bound and lower bound only match when $T=O(d^3H^3)$? If it does, how restrictive is this condition?’
>
> **A2**: We guess you want to say the upper and lower bounds match when $T = \Omega (d^3H^3)$, instead of $T = O (d^3H^3)$. Since $T = KH$, this suggests that we need $K = \Omega(d^3H^2)$ to make the upper and lower bounds match with each other. Since $d$ and $H$ are fixed parameters for the underlying MDP, this is a quite reasonable condition, as it basically requires the algorithm to interact with the environment sufficiently to achieve statistical optimality.

---

> > ### Author Response · Authors · 2021-08-30
> > **Follow-up Message**
> >
> > Thank you for your comments and suggestions! We believe our response has addressed all your comments and questions.
> > Please let us know if you have any other questions, and we are happy to discuss more. If you’re satisfied with our response, we sincerely hope you could reconsider the rating.

---

> > ### Comment · Reviewer_ZCnH · 2021-09-01
> > **Response**
> >
> > Thank you for your response to my reviews. I've updated my score accordingly.

---

> > > ### Author Response · Authors · 2021-09-02
> > > **Thank you for your positive feedback!**
> > >
> > > We are glad that our response has addressed your concerns! We will incorporate your comments in the final revision.

---

### Official Review · Reviewer_Xhcq · 2021-07-18

**Rating:** 6
**Confidence:** 4

**Summary:**

This paper proposes LSVI-UCB-Batch and LSVI-UCB-RareSwitch algorithms which can satisfy adaptivity constraints, i.e., the number of policy changes during learning is small. The authors proved regret bounds showing that with a small number of switches, the algorithms can achieve same regret bound as LSVI-UCB.

**Limitations And Societal Impact:**

I did not see many discussions on the limitations of this work. I did not see any negative societal impact.

**Main Review:**

Pros:

I think this paper is very well-written and easy to follow. The results are rigorously presented. I did not check the proofs but the results seem to be correct and aligned with prior works.

Cons:

I think the novelty of the algorithm design is relatively limited. The LSVI-UCB-Batch is essentially the same as LSVI-UCB with the only difference being that the policy is updated every K/B episodes rather than every episode. The determinant based criterion in LSVI-UCB-RareSwitch also appeared in several prior works.

The theory only applies to line MDP which is a bit limited in my opinion. But I understand that the recent line of work on provably efficient RL needs such assumption, especially for value-iteration type of algorithms.

Minor:

Line 10 in Algorithm 1: superscript “+” is not defined, similar thing for Algorithm 2.

Line 222:  “In detail, when B = T, LSVI-UCB-Batch degenerates to LSVI-UCB”: should it be T or K? I think by definition B cannot be larger than K. Similar things in lines 233-234.

**Time Spent Reviewing:**

3

---

> ### Author Response · Authors · 2021-08-10
> **Response to Reviewer Xhcq**
>
> **Q1**: 'Line 10 in Algorithm 1: superscript '+' is not defined, similar thing for Algorithm 2.'
>
> **A1**: Sorry for the undefined notation. $[a]^+$ denotes $\max(a,0)$. We will define it in the revision.
>
> ---
>
> **Q2**: ‘typo in line 222, 233-234’.
>
> **A2**: Thanks for pointing out the typos. It should be $B=K$ as $K$ is the total number of episodes. We will correct them in the revision.
>
> ---
>
> **Q3**: ‘I think the novelty of the algorithm design is relatively limited.’
>
> **A3**: We agree that the algorithms we proposed look similar to LSVI-UCB. Yet for the batch learning model, we think it is still valuable to analyze and verify whether RL algorithms can be adapted to uniform batch grids since this could be the actual situation in practice. As far as we know, this is the first algorithm proposed in RL to tackle such adaptivity issues. Also note that our work is the first to study RL with linear function approximation under adaptivity constraints, and the algorithms and results proposed in our paper are new.
>
> ---
>
> **Q4**: ‘The theory only applies to linear MDP which is a bit limited in my opinion.’
>
> **A4**: We agree with your point, yet we think the linear MDP is still important as a generalization of the tabular MDP. The key idea in the algorithm design and analysis developed in this paper is generic and can be generalized to other kinds of MDPs (e.g., linear mixture MDPs [Ayoub et al. 2020]). Our results take an initial step on adaptivity constraint problems in RL beyond the tabular setting and shed light on the algorithmic design and analysis under more general assumptions, e.g., RL with general function approximation [Wang et al. 2020].

---

> > ### Comment · Reviewer_Xhcq · 2021-08-29
> > **Thanks for your response**
> >
> > Thanks for your response. Overall I am positive about this paper. I'll keep my score.

---

> > > ### Author Response · Authors · 2021-08-29
> > > **Thank you for your positive feedback!**
> > >
> > > We will prepare the final version according to your comments.

---

### Decision · Program_Chairs · 2021-09-28

**Decision:**

Accept (Poster)

**Comment:**

This paper proposes LSVI-UCB-Batch and LSVI-UCB-RareSwitch algorithms which can satisfy adaptivity constraints, i.e., the number of policy changes during learning is small. The authors proved regret bounds showing that with a small number of switches, the algorithms can achieve same regret bound as LSVI-UCB. There is a consensus that the paper should be accepted.

**Consistency Experiment:**

NeurIPS has a long history of experimentation. In 2014, NeurIPS ran an experiment in which 10% of submissions were reviewed by two independent committees to quantify the randomness in the review process. This year, we repeated a variant of this experiment to see how the quality of the review process has changed over time.  This paper was part of the experiment and was therefore assigned to two committees (consisting of reviewers, an Area Chair, and a Senior Area Chair) that reached independent decisions.  If both committees made the same recommendation, this recommendation was followed. If a single committee recommended acceptance, the paper was accepted (with the exception of a few cases in which the other committee identified what we considered a fatal flaw, e.g., an error in a key result).

This copy’s committee reached the following decision: **Accept (Poster)**

The other committee assigned to the paper recommended **Reject**.  You can find the other set of reviews, along with any follow up discussion with the authors here:
https://openreview.net/forum?id=vYZmTEDFoqP